# mTBI-Induced Systemic Vascular Dysfunction in a Mouse mTBI Model

**DOI:** 10.3390/brainsci12020232

**Published:** 2022-02-08

**Authors:** Weizhen Lv, Zhuang Wang, Hanxue Wu, Weiheng Zhang, Jiaxi Xu, Xingjuan Chen

**Affiliations:** 1Institute of Medical Research, Northwestern Polytechnical University, Xi’an 710072, China; weizhenlv@mail.nwpu.edu.cn (W.L.); todd1997@mail.nwpu.edu.cn (Z.W.); zwh-original@mail.nwpu.edu.cn (W.Z.); 2Department of Physiology and Pathophysiology, Xi’an Jiaotong University Health Science Center, Xi’an 710061, China; wuhanxue@stu.xjtu.edu.cn (H.W.); xujiaxi@xjtu.edu.cn (J.X.)

**Keywords:** mTBI, cytokines, inflammation, aortic vasoconstriction and relaxation

## Abstract

Mild traumatic brain injury (mTBI) without skull fracturing is the most common occurrence of all TBIs and is considered as a serious public health concern. Animal models of mTBI are essential to investigation of TBI and its effects. In the current study, we developed and characterized a reproducible mouse model of mild TBI, meanwhile, the effects of this mTBI model, as well as repetitive mTBIs (rmTBIs), on the endothelial function of mouse aortas were also studied. In variety of closed-head models of mTBI, impact velocity, weight, and dwell time are the main parameters that affect the severities of injury. Here, we used a device, converting parameters of velocity, tip weight, and dwell time into impact force, to develop a mouse model of close-head mTBI. Mice were subjected to a mild TBI induced by the impact forces of 500, 600, 700 and 800 kdyn, respectively. Later, brain injuries were assessed histologically and molecularly. Systemic and brain inflammation were measured by plasma cytokine assay and glial fibrillary acidic protein (GFAP) staining. The composite neurobehavioral test revealed significant acute functional deficits in mice after mTBI, corresponding to the degree of injury. Mice brain undergoing mTBI had significant elevated GFAP staining. Plasma cytokines interleukin-1β (IL-1β) and superoxide dismutase (SOD) were significantly increased within 2 h after mTBI. Taken together, these data suggest that the mTBI mouse model introduce within our study exhibits good repeatability and comparable pathological characters. Moreover, we used this mTBI mouse model to determine the effect of single or rmTBIs on systemic vasoconstriction and relaxation. The isometric-tension results indicate that rmTBIs induce a pronounced and long-lasting endothelial dysfunction in mouse aorta.

## 1. Introduction

Mild traumatic brain injury (mTBI) is one of the most common brain injuries, effecting millions of people worldwide. In the United States, mTBI approximately occurs in 1.6–3.8 million cases annually [1]. When an external physical force impacts the head, TBI occurs, damaging the brain by fracturing the skull or causing the brain to move within the intact skull [2]. Non-penetrating physical impacts can cause an mTBI without skull fracture with characters of short deterioration of neural function that may or may not involve loss of consciousness [3,4]. mTBI is less frequently involved in hospital stays and follow-ups due to the minimal symptoms after a mTBI. Generally, a head injury is classified as a mTBI according to the following criteria: normal structural imaging; loss of consciousness <30 min; alteration of consciousness less than 24 h; post-traumatic amnesia of less than a day; and an initial Glasgow Coma Scale of 13–15 [5]. However, mTBI results in a complex cascade of injuries in brain [6]. It has become clear that repeated mTBIs can induce permanent dysfunction, including neurodegenerative disorders [7,8] and cardiovascular complications [9,10], therefore bringing a substantial burden to patients, families, and the public health system [11,12,13].

Although mTBI has been associated with a broad of long-term problems, there is still a lack of knowledge regarding the pathophysiological impacts of mTBI. The animal models of mTBI are helpful to the investigation of mTBI-associated pathologic changes, and provide opportunities for diagnosis and treatment of mTBI in clinic. However, it is impossible to simulate every aspect of TBI in a single animal model. Currently, several TBI models have being used in animals of various ages and injury severity levels. Among them, rodent models are the most common in TBI researches, due to research ethical reasons, their lower costs and small size. Several research groups have developed the closed head injury or concussive head injury (CHI) mouse model that provide a method to investigate mTBI in an efficient, rigorous, and controlled manner [14,15]. Optimal mTBI model should fulfill the requirements that are at a mild level of severity, closed, with low mortality rate and exhibit no obvious symptoms of neuropathy [2]. Within those requirements, the severity of experimental injury is crucial in modeling mild TBI, and the levels of injury in most mTBI models are controlled from mild to severe by changing the injury parameters including impact velocity, weight, and dwell time.

Impact velocity, weight and dwell time determine the impact force on TBI animals. In the current study, we developed a model of closed skull mTBI in mice and the level of injury is controlled by electromagnetic impact force. The surviving mTBI mice without skull rupture showed significant brain and systemic inflammation with or without acute neuromotor deficits. Furthermore, we used this mTBI model to investigate the acute and long-term effects of single mTBI or repetitive mTBI on aortic constriction and relaxation utilizing isometric tension measurements.

## 2. Materials and Methods

### 2.1. Animal

C57/BL6 male mice (8–10 weeks of age) were purchased from Beijing Vital River Laboratory Animal Technology Co., Ltd. (Beijing, China) and/or were bred in house. All animal procedures were approved by the Northwestern Polytechnical University Medical and Experimental Animal Ethics Committee (No. 2019029). The mice were euthanized under isoflurane anesthesia by decapitations.

### 2.2. In Vivo Model of Mild Traumatic Brain Injury

Before Impact-The composite neuroscore test (CN) was implemented to evaluate pretraumatic neurological function [16]. Mice underwent a battery of tests: right and left forelimb and hindlimb flexion and right and left lateral pulsion. For each task, animals were scored on a scale of 0–4, with 0 valued at severely impaired and 4 at no impairment. Mice were anesthetized with 3% isoflurane followed by maintenance of anesthesia with 2% isoflurane. The level of anesthesia was monitored until the animal reached the level of surgical anesthesia, characterized by showing loss of pedal withdrawal reflex. The mice were placed prone on the heating pad to keep the body temperature maintained at 37 ± 0.5 °C during the operation. A funnel-shaped nasal cone was used to keep the mice under anesthesia. The head top was shaved and a mid-line incision was made from bregma to lambda, to expose the surface of the skull. The Icare TONOLAB (Vantaa, Finland) was used to measure the intraocular pressure before impact (Figure 1A).

Impact-A mouse was placed on the stereotactic positioning device of the electric brain injury impact instrument (Beijing Zhongshi Technology Company, ZS-FD/NL). The mouse’s mouth was placed on the bite bar with the front teeth into a hole, and the anesthesia cone covered the nose of the mouse to maintain anesthesia. Gently placed the two ear bars into the ear canal and secured and make sure that ear bars were at the same height as the bite bar (Figure 1B). A manipulator was used to slowly adjust the position of the rammer to ensure that the rammer just touched the mouse brain and was located 1/3 of the way between the ear canals, then set this position to zero. Adjusted the vertical height of the rammer according to the depth of the impact required: 2 mm and 3 mm. Head injury was induced by an impacting piston with a tip diameter of 2.5 mm. Sham-injury mice were anesthetized with isoflurane, but were not subjected to the head. Adjusted force parameters of accumulator to meet different impact force requirements. Pressed the switch, the rammer fell and completed a brain impaction. Brain injuries were introduced at four different impact forces (kdyn): 500, 600, 700 and 800. The intraocular pressures were also recorded after every impaction.

Post Impact-Placed the animals on a warm pad to maintain their bodies’ temperature. Do not leave the animals unattended until they regained sufficient consciousness. The composite neuroscore test above were carried out when the animals were fully conscious. Two hours after the impaction, the animals were anaesthetized deeply and their blood was collected into anticoagulant tubes by removing eyeball. They were then immediately decapitated; the brain tissues and the thoracic aortas were collected for future assays.

### 2.3. Histological Detection

The brain tissues were prepared as described above. Isolated mice’s brains were fixed in 4% paraformaldehyde overnight, and then embedded in paraffin. Paraffin blocks were cut into 5 μm-thick sections, which were de-paraffinized and rehydrated after mounting on glass slides. Hematoxylin and eosin (H&E) staining was performed using the H and E dye solution set (Servicebio, Wuhan, China, G1003). Sections were examined using the Digital Slide Scanner (Winmedic, Shandong, China). Immunostaining was performed as following: samples were permeabilized in 0.2% Triton X-100 in TBS for 5 min and were then washed three times in TBS. After blocking in 5% horse serum in TBS at room temperature for 1 h, slides were incubated with anti-GFAP (1:1000, Servicebio, GB12096) overnight at 4 °C, placed in a wet box. GFAP immunome activity was detected with Cy3 conjugated Goat Anti-mouse IgG (1:4000; Servicebio, GB27301). Stained samples were mounted with Antifade Mounting Medium with DAPI (Beyotime, Nantong, China, P0131) and visualized by confocal microscopy (Olympus Fluoview FV1000).

### 2.4. Isometric Tension Measurement

Mouse aortas were isolated and cleaned from the fat and connective tissues in PSS solution. Vessel isometric tensions were measured using the Multi Myograph System-model 620M (DMT, Denmark). Briefly, aortic arches were cut into 2–3 mm rings, then hung on the wires of myograph and were placed into the 5-mL tissue baths filled with the standard PSS buffer maintained at 37 °C and continuously oxygenated by bubbling a gas mixture of 95% O_2_ and 5% CO_2_ during all of the performed experiments. The preload in all experiments was set to 0.7–1  mN. Increasing concentrations of phenylephrine and then acetylcholine was added directly into the tissue baths while the contraction force was measured. SNAP (S-Nitroso-N-acetyl-DL-penicillamine, a nitric oxide donor) was used to assess the maximal receptor-independent aortic ring dilations. The analog ring tension data was then digitized and recorded on a computer’s hard drive.

### 2.5. Drugs and Solutions

All drugs were purchased from Selleck or Macklin. The solution composition was as follows. The standard PSS buffer contained (in mM): 130 NaCl, 4.7 KCl, 1.6 CaCl_2_, 1.18 NaH_2_PO_4_, 1.17 MgCl_2_, 14.9 NaHCO_3_, 5.5 glucose and 0.026 EDTA. The 70 mM KCl solution contained (in mM): 74.7 NaCl, 60 KCl, 1.6 CaCl_2_, 1.18 NaH_2_PO_4_, 1.17 MgCl_2_, 14.9 NaHCO_3_, 5.5 glucose and 0.026 EDTA.

### 2.6. Statistical Analysis

The Sigma Plot12.5 and GraphPad Prism 6 were used to analyze the data. The unpaired t test was utilized to determine whether there is a statistically significant difference between the two data sets with normally distributed populations and equal variances. The Two-way ANOVA test followed by the Student-Newman-Keuls post hoc all pairwise multiple comparison test were used to compare among the experimental groups affected by two different factors when the data sets were normally distributed populations with equal variances. The data were presented as mean ± standard error and were considered significantly different if the *p* value was <0.05.

## 3. Results

### 3.1. The Impact Force in mTBI Model

In the current study, the level of brain injury was controlled by changing the impact force. Therefore, the effect of impact force on injury severity was firstly addressed. Fully anesthetized male mice were subjected to a single electromagnetic impact force with a range on impact force from 500 to 800 kdyn in 100 kdyn increments at 2 mm depth. We measured intraocular pressure (IOP) changes in the mouse eye with the Icare TONOLAB (Vantaa, Finland) before and after the injury within 30 min, to confirm the severity of the cortical bone impact and the absence of skull fracture. All mice recovered well from the injury in impact force of 500 and 600 kdyn groups and were able to ambulate within 30 min of injury. There were no significant IOP changes in these two impact conditions (Table 1). One in five mice died after 700 kdyn impact and no IOP was detected. As the impact force reached 800 kdyn, more than half of mice died during the injury. When the depth of impact reached 3 mm, only one in five mice surviving even in 600 kdyn impact injury (data not shown). Mice surviving from injuries showed unchanged IOP (Table 1), suggesting the absence of skull fracture. Therefore, the model with the impact parameters of 500 and 600 kdyn and 2 mm depth could fulfill the mTBI model requirements with head-closed and low mortality.

### 3.2. Histological and Behavioral Characteristics of the Injury

We then evaluated whether the models introduced histological alterations in brain of the injured mice. Animals subjected to mTBI with different impact forces were euthanized 2 h after injury and the brains were fixed for hematoxylin and eosin (HE) and GFAP staining. HE staining of brains from mice with different groups of mTBI showed virtually little to no histological evidence of injury (Figure 2A). There was no histological evidence of injury to cortex or hippocampus in mice in the sham group (Figure 2A). Since no detectable histopathologic alteration was present in all TBI groups, further evaluation with glial fibrillary acidic protein (GFAP), a marker of brain injury, were performed. The staining of GFAP in hippocampus was significantly increased after injury in an impact force dependent manner (Figure 2B).

To evaluate systemic inflammation, we then analyzed 3 types of inflammatory cytokines, including IL-6, IL-10, and IL-1β, in the plasma of the sham and mTBI mice. At 2 h after injury, the level of IL-1β (pg/mL) in plasma increased significantly (Figure 3B, sham: 1037 ± 45 vs. 1421 ± 101 (500 kdyn), 1364 ± 47 (600 kdyn), 1405 ± 90 (700 kdyn)), while the level of IL-10 was not significantly changed, in all mTBI groups compared to the sham group, and this is consistent with the TBI data obtained in human subjects [17] and other reported rodent mTBI models [15,18]. Increased IL-6 level was another important biomarker in human mTBI [17], as well as in rodent mTBI models [15,18]. However, we failed to detect significant change in the plasma level of IL-6 in our mTBI mouse model, compared to the sham group. After TBI, assembly of oxidative stress-related molecules are produced in the brain and the innate antioxidants, such as SOD, are activated accordingly [19]. In this case, we measured the plasma SOD level using an ELISA kit, and it was showed in the results that the plasma SOD levels (ng/mL) of the mTBI mice were also elevated with the impact forces of 500 and 600 kdyn (Figure 3A, 13.6 ± 0.7, *n* = 5 and 13.0 ± 0.4, *n* = 5, respectively), compared to the sham group (10.34 ± 0.46). However, the SOD level was not significantly changed in the mTBI mouse with 700 kdyn impact force (10.61 ± 0.42).

Immediately following the head injury with different impact forces, hindlimb and forelimb flexion and lateral pulsion task were performed to evaluate the post-traumatic neurological deficits. Although mTBI with 500 kdyn impact force did induce significant acute neuroinflammation after injury, for hindlimb and forelimb flexion, only the mice with TBI impacts of 600, 700 and 800 kdyn, but not 500 kdyn, showed markedly lower scores during the test (Figure 4). Similarly, the results of lateral pulsion task test showed that TBI mice with 600, 700 and 800 kdyn impacts, but not 500 kdyn, exhibited behavioral impairment, compared to the concurrently tested, sham-injured mice.

### 3.3. Effect of mTBI or Repeated mTBI on Aortic Constriction and Dilation

TBI effects predominantly the brain, but it is also known to cause impairments in other organs [20,21]. There is evidence suggesting that TBIs have significant effect on the systemic vasculature, including mesenteric and aortic vasculature [9,10]. Therefore, we further investigated the possible whether this mTBI causes endothelial dysfunction in the conduit systemic circulation-induced our mTBI modeling. Firstly, isometric tension recordings were performed on rings of aortic arch obtained from mice subjected to the mTBI with 600 kdyn impact force or sham surgical procedure after 2 h post-injury. As shown in Figure 5, mTBI showed no effect on the constructions induced by 70 KCl, while, it increased the amplitude of phenylephrine (PE)-induced contractions of the mTBI rings, compared to sham rings (10 μM PE induced active tension normalized to the peak amplitude of 70 mM KCl-stimulated contraction: 0.56 ± 0.1 versus 0.81 ± 0.2 for sham and mTBI, respectively, Figure 5), indicating that the mTBI aortas were more sensitive to phenylephrine. We then assessed endothelial function by examining the acetylcholine (ACh)-induced relaxation of aortic rings, which pre-contracted with phenylephrine. Compared to sham rings, the mTBI rings exhibited significantly reduced acetylcholine-induced dilations (10-μM ACh-induced dilations: 88.59 ± 13.2%, *n* = 5 versus 45.9 ± 18.9%, *n* = 6 for sham and TBI, respectively, Figure 5). We also investigated whether mTBI-induced changes in vascular reactivity and endothelial function would persist for a longer period, and no significant difference was detected between the TBI and sham groups 21-day post-injury (Figure 6A,B), which was consistent with the observations in previously reported mTBI mouse model [9].

It is clear now that rmTBI occurring within a short period might be devastating or fatal [22,23]. We also investigated whether rmTBI-induced changes in vascular reactivity and endothelial function would persist for a longer period. Animals were subjected to mTBI (600 kdyn impact force) three times every 72 h to generate an rmTBI model. At 21st day post last injury, mice were sacrificed and the aortas were isolated to perform the isometric experiment as described above. The amplitude of phenylephrine-induced contractions of the aortic rings was not markedly changed, (Figure 6C) in both mTBI and rmTBI models, compared to the sham rings. However, the rmTBI rings exhibited significantly reduced acetylcholine-induced dilations (74.2 ± 8.1%, *n* = 3 versus 51.9 ± 6.1%, *n* = 6 for sham and rmTBI, respectively). Therefore, the endothelia dysfunction caused by rmTBI in our hand did last for a longer period of time.

## 4. Discussion

In comparison to the animal models of moderate-to-severe TBI, the models of mild TBI are less studied. With the alarming number of people, especially adolescence, suffering from mTBI, and with the realization that mTBI may exhibit long-lasting negative impacts on the health of patients [13], a variety of mTBI rodent models have been developed. Herein, we described a mouse model of mTBI and the level of injury severity was controlled by impact force, which is unique from previous reports. This mTBI model can be utilized for both single impact and repetitive impact in a short time. We further investigated the effects of single or rmTBI on aortic constriction and relaxation. It was found that mTBI induced an increased constriction and decreased dilation 22 h after the injury. In our case, the rmTBI was able to cause long-lasting systemic endothelial dysfunction.

Bodnar et al. summarized the so-far reported rodent mTBI models [14], and the major body is still weight drop models, followed by piston-driven models. Technically, electromagnetic driven piston are the most common used devices and the device is able to produce consistent, graded CCI injuries in rodent without the need for frequent calibration [24]. Injuries caused by a piston are typically induced by zeroing the piston on the surface of the skull or scalp and then delivering an injury at a specific depth, velocity, or impact force. The impact velocity, impactor tip size and dwell time in the reported mTBI models were actually various [14]. Moreover, those parameters can be converted to impact force according to the mechanical formula. The parameter of the device we used here to induce injury was the impact force, which combined impact velocity, weight and dwell time. We further investigated the injury severity with the increased impact force by monitor the behaviors and cytokine of the mTBI mice. The result indicated that impact force of 500 and 600 kdyn increased GFAP staining in brain and the serum level of IL-1β and SOD without mortality. When the impact force increased to 700 kdyn, we started to lose animals due the rupture of the skull. Therefore, the model with the impact parameters of 500 and 600 kdyn and 2 mm depth may fulfill for the mTBI model requirements of head-closed and low mortality. The model is capable of producing a wide range of injury severities, as assessed histologically and behaviorally; there was a clear gradation in both brain inflammation and acute behavioral deficits in adult wild-type mice as the impact force was increased from 500 to 700 kdyn. That would give a clue for the further mTBI model developing with the parameter of tips weight, impact velocity and dwell time.

The severe mTBI may induce parenchymal hemorrhage after impact [15]. In contrast, our mTBI model had no evidence of histological injury according to the HE staining. However, the GFAP staining indicated the activation of astrocytes, which are activated in response to central nervous system (CNS) injury or exogenous substances intervention. Multiple studies supported the notion that astrocytes play a key role in the pathogenesis of TBI, including promotion and restriction of neurogenesis and synaptogenesis, acceleration and suppression of neuro-inflammation, and disruption and repair of the BBB via multiple bioactive factors (see reviews [25,26]). Accumulate evidence indicate that microglial activation has also become gradually recognized as a key process regulating the pathology in TBI [27,28,29]. However, post-mTBI microglial activation is highly time-sensitive and the microglial activation in mTBI animal models is non-standardized. That limit us to identify actual patterns of post-mTBI microglial activation over time. We also observed the animal behavioral deficits after the mTBI. However, more investigations are needed to clarify the relationship between the behavioral deficits and the activation of astrocytes.

Increased level of blood-based pro-inflammatory cytokines is linked to acute TBIs [30]. IL-6 is a promising biomarker cytokine of brain injury in mTBI patients [17] and mTBI rodent models [15]. However, in the current study we failed to detect significant change in the level of serum IL-6 levels after mTBI modeling using ELISA method. Since the serum was collected 2 h post-injury, it cannot be excluded that the time is too short to observe the increased IL-6 level. In TBI, pro-inflammatory microglial activation is usually accompanied by an increase in the pro-inflammatory cytokines’ synthesis, and the release of NO, ROS and free radicals. SOD performs protective functions in the acute period after injury [31,32]. In our study, elevated serum SOD was found in mTBI with 500 and 600 kdyn, but not 700 kdyn. The SOD level may correlate with the severity of the injury. In our model, mTBI mice with 700 kdyn impact showed more serious posttraumatic neurological deficits. Actually, another mouse mTBI model weight drops with cognitive impairments also showed unchanged endogenous SOD level [33]. mTBI with 500 or 600 kdyn impact showed no or less posttraumatic neurological deficits may relate to the protection of increased SOD level.

We have reported that mTBI closed-head mild TBI causes long-lasting systemic endothelial dysfunction, which is resolved between 7- and 21-day post-injury, in mice by overacting TRPC6 channels [9]. At 21-day post injury by single impact, the dysfunction of endothelium cannot be detected anymore. It is known that repetitive mTBI can affect brain or other organs for a quite long period of time. Therefore, we developed repetitive mTBI model with 600 kdyn impact force at 2 mm depth. In addition, the endothelial function was significant decreased 2 h after single mTBI but not 21 days post injury. Our previous study indicated that the endothelial dysfunction induced by TBI might be due to the overacting of TRPC6 channels, which could be activated by some increased levels of chemokine through TLR4. Recently, Sackheim et al. reported that TBI-induced impaired endothelial dysfunction might due to the disruption of inward-rectifier potassium channels [34]. However, the molecular mechanism underlying the endothelial dysfunction induced by rmTBI is still unclear.

## 5. Conclusions

Here, we described a closed head injury model of mild TBI that showed repeatable mild injury in mice by controlling the force of impact. We observed significant changes in brain and systematic inflammation that correlate to the severity of injury. The neurological deficits of mTBI mice appeared with the increase of the impact force. The model we describe may serve as a valuable reference for further mTBI modeling. Furthermore, we found that rmTBI may results in long lasting dysfunction of systematic endothelium.

## Figures and Tables

**Figure 1 brainsci-12-00232-f001:**
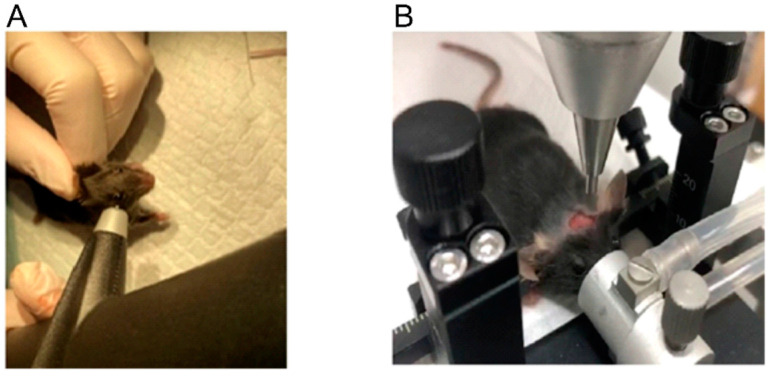
IOP measurement in mice (**A**). The placement of mice before impact (**B**).

**Figure 2 brainsci-12-00232-f002:**
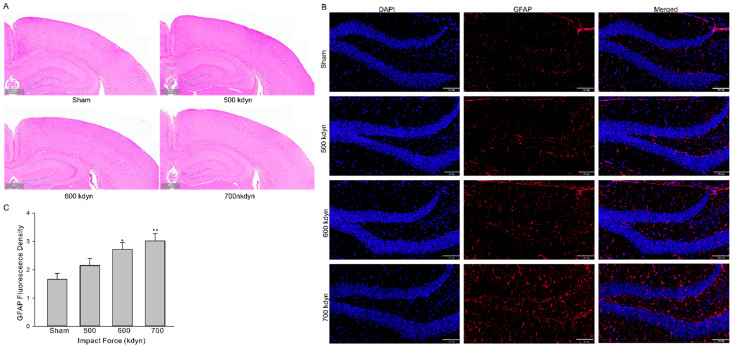
Histological analysis of the brain in the frontal cortex and hippocampus 2 h after mTBI with various impact forces. Sections were stained with hematoxylin and eosin (**A**) or GFAP (**B**). Scale bars indicate 500 μm in (**A**) and 100 μm in (**B**). (**C**) Summary data for (**B**). Data are presented as mean ± SEM. * *p* < 0.05, ** *p* < 0.01 compared to sham.

**Figure 3 brainsci-12-00232-f003:**
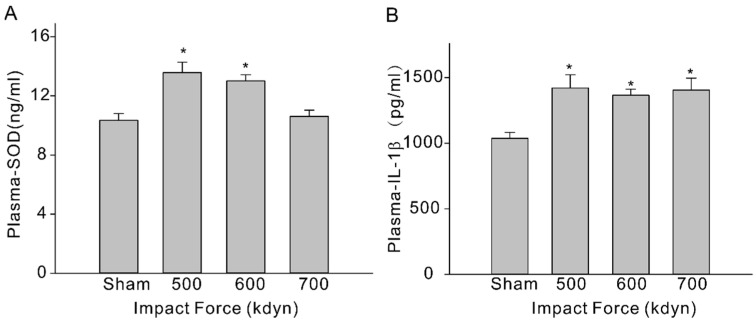
Plasma SOD (**A**) and IL-1β (**B**) levels in sham and mTBI mice. Plasma were collected at 2-h time-point after the impact, and samples were analyzed via ELISA. Data are presented as mean ± SEM. * *p* < 0.05 compared to sham.

**Figure 4 brainsci-12-00232-f004:**
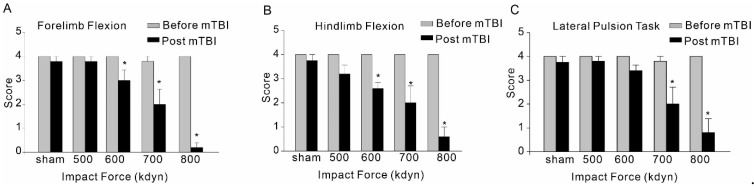
Effect of mTBI caused by gradient impact force on composite neuroscore behavioral, including forelimb flexion (**A**), hindlimb flexion (**B**) and lateral pulsion task (**C**), in C57/BL6 mice. Testing was completed within 2 h after TBIs. The dead mice scored zero. Data are mean ± SEM. * *p* < 0.05 compared to sham.

**Figure 5 brainsci-12-00232-f005:**
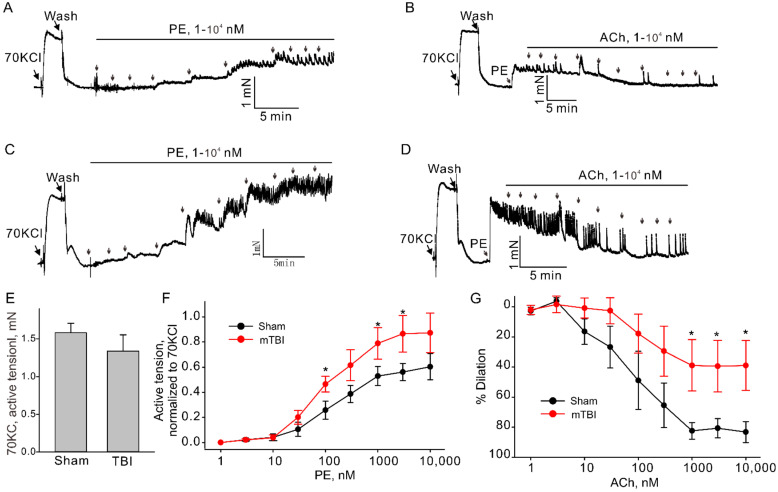
Effect of mTBI on endothelial function in mouse aortic arches 2 h after mTBI. (**A**,**C**). The representative traces of active tension changes in sham and mTBI mice induced by phenylephrine (PE). (**B**,**D**). The representative traces of aortic dilation induced by acetylcholine (ACh) in sham and mTBI mice. (**E**). Statistical graphs comparing the active tensions-induced by 70 mM KCl. (**F**). Concentration-response relationships for PE-induced contractions in aortic arch rings from sham and mTBI mice. (**G**). Concentration-response relationships for Ach-induced dilation. Data are presented as mean ± SEM, * *p* < 0.05 compared to sham.

**Figure 6 brainsci-12-00232-f006:**
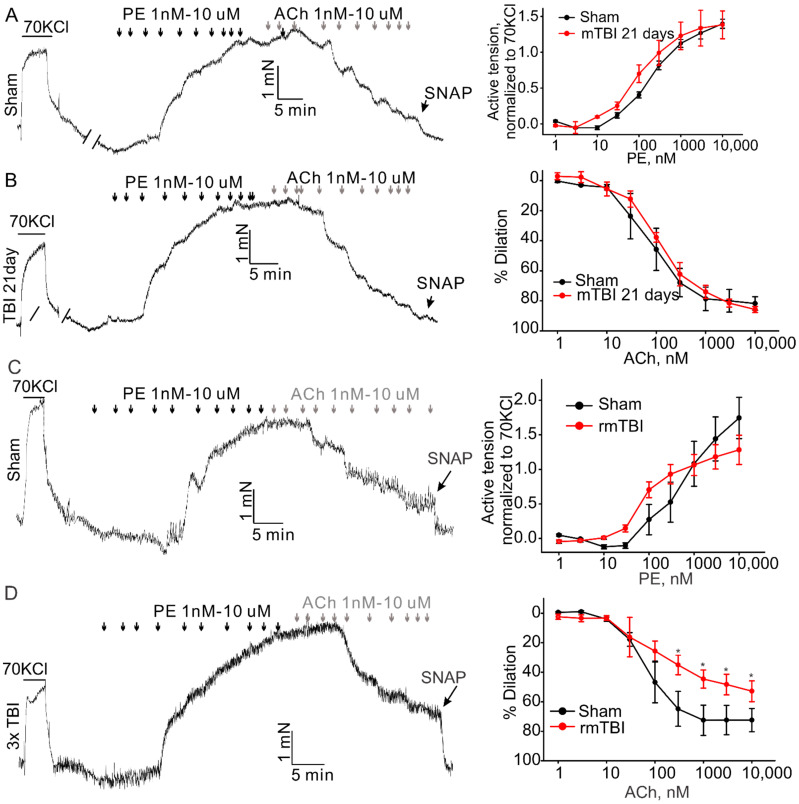
Effect of mTBI or rmTBI on aortic endothelial function 21-day post injury. The left column shows the representative traces of active tension changes in sham (**A**,**C**) or mTBI (**B**,**D**) mice in the present of phenylephrine and followed by acetylcholine. The right column shows concentration-response relationships for phenylephrine (PE)-induced contractions and ACh induced dilations. PE-phenylephrine, ACh-acetylcholine. Data are presented as mean ± SEM * *p* < 0.05 compared to sham.

**Table 1 brainsci-12-00232-t001:** Intra-ocular pressure.

Impact (kdyn)	Intra-Ocular Pressure (mmHg)		Survival
	Before Impact	After Impact	
500	9.5 ± 0.5 (*n* = 6)	8.8 ± 0.7 (*n* = 6)	100%
600	9.6 ± 0.6 (*n* = 6)	8.8 ± 1.2 (*n* = 6)	100%
700	9.6 ± 0.4 (*n* = 6)	10.6 ± 1.4 (*n* = 5) ^#^	83.3%
800	9.0 ± 0.48 (*n* = 5)	9.0, 7.0 *	40%

^#^ Five of six mice survived under this experimental condition. * Two of five mice survived under this experimental condition.

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
