# Peer review of "mTBI-Induced Systemic Vascular Dysfunction in a Mouse mTBI Model"

_brainsci, 2022, doi:10.3390/brainsci12020232_

Round 1
Reviewer 1 Report
1. Authors claimed that there are no differences in hematoxylin and eosin staining which indicates that there is no neuronal loss although their TBI mouse model showed robust astrocytes activation and behavioral deficits. How authors explain that behavioral phenotype was mainly by activation of astrocytes and there is no involvement of neurons in these deficits?
2. It would be interesting to check microglia activations in these mouse models to determine ethe microglia contribution in these observed phenotypes.
3. For a better conclusive results, n needs to be increased.
4. Discussion needs to be written well to clarify the concepts the authors trying to point out.
5. Overall, manuscript needs proofreading to improve the quality of manuscript.
Reviewer 2 Report
A mouse mTBI model and mTBI-induced systemic vascular dysfunction
The work is of importance to the TBI research however the written language needs significant improvement without which the manuscript seems incomprehensible.
Abstract should be restricted to add hypothesis before results for all the hypothesis tested in the current study. This will help the readers navigate the manuscript easily.
Introduction: Line 31, please provide accurate Statistics on mild TBI incidence, the current line sounds vague.
Line 52-53 sounds like results rather than hypothesis in introduction.
Please provide an image of the injury model with mice placed on it before injury, since it will help the readers to understand the procedure fully. Was any foam padding used underneath the animal? What was the material of the foam pad? This affects how the injury is impacted by the animal.
Line 94, do the authors know blood can be withdrawn without removing eyeballs? This is just a suggestion for future studies and can be used for repeated blood sample without significant damage to the animal.
GFAP levels seems to increase as soon as 2 hours after injury at this modest level of injury. It seems unusual in similar severity of injury. It would be interesting and affirming to see microglial expression at this timepoint as well.
How was the cortex impacted by mild injury in terms of GFAP expression?
Quantitative analysis of GFAP and microglial expression is needed.
Line 154: Authors state only upto 600kdyn to be mild TBI however they discuss 700 and 800 kdyn as mild TBI in figure 3, please clarify this.
Reviewer 3 Report
Title: A somewhat better title might be " A mouse mTBI model of mTBI-induced systemic vascular dysfunction".
Abstract: Abstract is clearly written.
Introduction: The introduction lacks a clear definition of mTBI in humans as well as criteria for a good mouse model of mTBI. Many of the sentences in the introduction are not sentences but sentence fragments lacking either subjects or verbs.
Some word errors occur (e.g. neuropathy for neuropathology) as do error in word inflection (survived for surviving). Careful editing and correction is needed.
Methods:
Authors should carefully state whether their method is a weight drop or piston model.
I prefer "impact" as in "before impact" and "post impact" in place of "impaction".
The parameters of the model have not been fully defined: dwell time, weight, velocity, depth, force etc.
Sentence fragments such as "Observed the ruler to ensure..." need to be written as sentences.
What is the relation between F and distance in determining Work? What determines injury? Force? Work? How do Force, depth, and dwell time contribute to extent of injury?
Results: The number of mice studied seems small. Any reason for the small N.
Sections 3.3 ande 3.4 need re-writing for clarity.
Discussion:
When referring to reference use only first name of authors (e.g. Bodnar et al.; Sackheim et al.)
Discuss how the proposed model meets the criteria for an animal model of mTBI.
The discussion would benefit from careful re-writing for clarity and grammatical correctness.
I personally use the grammar and spell checker in Word for my documents. The program Grammarly is also useful to catch writing errors. However, the assistance of a skilled English editor would be helpful in revising this paper.
Round 2
Reviewer 2 Report
HI, thank you for revising the manuscript. Please add results of microglial staining in the main manuscript with the figures. The poor quality of the staining should be acknowledged in the limitations section.
Reviewer 3 Report
Most of my concerns about this paper have been addressed by careful editing of this paper which now reads much better and I favor publication.
Not all of the references cited are correctly formatted. It appears that the authors have written the paper using a Latex editor such as Overleaf. If the references were downloaded accurately from Google Scholar or another authoritative source, the formatting of the references should be automatic if the correct style was selected. Brains Sciences guidelines suggest the ACS format for references. I recognize that authors are not primary English speakers but it is important that references be correctly formatted as LASTNAME then initials of FIRST and MIDDLE names for ACS style.
